# The Risk Factors of Chronic Pain in Victims of Violence: A Scoping Review

**DOI:** 10.3390/healthcare11172421

**Published:** 2023-08-29

**Authors:** Allison Uvelli, Cristina Duranti, Giulia Salvo, Anna Coluccia, Giacomo Gualtieri, Fabio Ferretti

**Affiliations:** 1Department of Medical Science, Surgery, and Neurosciences, University of Siena, Viale Bracci, 53100 Siena, Italy; 2Azienda Ospedaliero-Universitaria Senese (AOUS), Viale Bracci, 53100 Siena, Italy

**Keywords:** risk factors, victims of violence, chronic pain, health, scoping review

## Abstract

Violent situations are unfortunately very frequent in women and children all over the world. These experiences have long-term consequences for adult physical and psychological health. One of the most reported is chronic pain, defined in various sub-diagnoses and present in all types of violence. Unfortunately, the etiology of this condition is not clear and neither are the predisposing factors. The aim of this scoping review is to examine the literature trends about the probable risk factors of chronic pain in violence victims. Considering a bio-psycho-social model, it is possible to hypothesize the presence of all these aspects. The results will be discussed in the present article.

## 1. Introduction

Childhood abuse and intimate partner violence (IPV) are two major global public health problems with serious adverse health consequences across life [1,2,3]. Every year, about 4–16% of children are physically abused and one in ten is neglected or psychologically abused [4]. Sexual abuse is experienced by 15–30% of girls and 5–15% of boys [4], and IPV is experienced by 35% of women [5]. The abuse involves all types of violence: sexual, physical, and psychological. Child maltreatment is more likely to occur in families afflicted with domestic violence and is associated with re-victimization in adulthood [6,7].

Evidence suggests that these experiences have long-term consequences on adult physical and psychological health [8,9]. The primary outcome is the onset of chronic pain [10], understood as the type of pain or discomfort in the involved area that tends to persist for at least 3 of the past 6 months. It is reported by 48% [11] to 84% [12] of abused women. The most common diagnosis of victimized are pelvic pain [13], fibromyalgia [14], irritable bowel syndrome/bowel symptoms [15], abdominal pain [16], migraine/headache [17], back pain [18], chest pain [19], and neck pain [20], which tend to become chronic as well as the pain derived from them. Unfortunately, the etiology of these conditions is not clear. 

Previous studies found a relationship between chronic pain and specific psychological aspects [21], and neuroimaging studies showed that the activated brain areas by nociceptive stimuli are the same ones involved in emotional and behavioral states [22]. According to Garcia-Larrea and Peyron [23], pain elaboration involves three levels of neural connections. The first level processes the nociceptive activation of the spinothalamic tract. In the second level, the nociceptive stimulus is processed by the anterior cingulate cortex (ACC), insula, prefrontal cortex (PFC), and posterior parietal cortex, thanks to which the stimulus is consciously perceived, cognitively and attentionally modulated, and transformed into somatic or vegetative responses. The reappraisal of the stimulus takes place through the emotional context and individualized psychological factors that influence memory formation. The third level includes the orbitofrontal, perigenual ACC, and anterolateral PFC regions that can lead to inhibition or increase in the nociceptive stimulus perceived. 

Mood disorders (depressive disorder, dysthymia, and bipolar disorder) are often present in chronic pain patients, ranging from 1% to 61% [24,25,26]. People affected by chronic pain were 2.0 to 2.5 times more likely to experience an episode of depression at 6- and 12-month follow-ups than individuals without chronic pain [27,28], and pain-free individuals with depressive disorder were 4 times more likely to develop chronic pain at 6- and 12 months follow-ups than not depressed individuals [29,30]. The cause–effect relationship would appear to be complicated and bidirectional. The same results were obtained for all anxiety disorders, whereas the prevalence in chronic pain patients ranged from 1% to 50%, with the same bidirectional relationship [31,32,33]. Also, for alcohol, drugs, and smoking, there is the same effect, with a prevalence of 1% to 25% [34,35,36]. Furthermore, suicidal thoughts are present in 28–48% of the cases [37,38], and violence doubles the odds of chronic pain onset [10]. 

Chronic pain, as opposed to acute pain, would seem to have little to do with injuries, even if it is still associated with biological processes; in fact, an interaction of sensory, autonomic, endocrine, and immune responses contribute to the nociceptive stimuli perceived [39]. The nervous system plays a role by detecting threats, signaling dangers, and starting a response to them, the endocrine system causes an arousal response to increase the survival odds thanks to the stress response, and the immune system detects microbial invasion and toxins and initiates complex inflammatory responses [40]. These processes could collectively compromise a defensive biological response to pain. Pain can persist as a focus of chronically disorganized, locally inflamed processes that respond maladaptively to systemic changes at the nervous, endocrine, and immune levels. In many chronic cases, the local tissue environment appears to repair itself, but sensory processes remain abnormal, creating chronic pain [39]. Trauma, such as violence and abuse, can cause the same biological reactions to nociceptive stimuli, activating the autonomic nervous system, which triggers the immune system and inflammatory response [41]. Inflammation can increase the risk of psychopathology by altering the metabolism of neurotransmitters, and psychopathology can similarly increase the risk of chronic pain [42]. 

Considering the complexity of the examined situation, the bio-psycho-social model [43] could help us to understand the phenomenon better. According to this model, the disease results from multiple variables: biological, psychological, and social, that co-occur in different ways for each person. Therefore, considering this approach, we might expect that the chronic pain in violence victims is not attributed exclusively to a single cause, psychological, sociological, or biological, but to a combination of the three. It remains to be understood in which way and prevalence these three categories are present in abused women. 

This scoping review aims to analyze the research trends about the bio-psycho-social components most associated with chronic pain in victims of violence, considering all abuse types and the primary chronic pain diagnoses in violence cases. It is crucial to know them because if patients showed the major risk factors, preventive measures could be taken to counter the onset and/or chronicity of pain. This type of review does not exist in the literature, and thanks to it, it will be possible to increase the knowledge about this condition and help the victims from a clinical and research point of view. It would be possible to offer new solutions and therapeutic strategies from a clinical point of view and to orient studies about the creation of a new tool to evaluate the presence of these aspects.

## 2. Materials and Methods

This search protocol was based on the Preferred Reporting Items for Systematic Reviews and Meta-Analysis extension for Scoping Reviews (PRISMA-ScR) guidelines [44], according to the PECOS (Population, Exposure, Comparison, Outcome, Study Design) guidelines.

### 2.1. Search Strategy

The research was conducted on the electronic databases of PubMed, Scopus, Web of Science, and ERIC from March 2023 to June 2023 and we carried out a manual review of references. The search strategy relating to the risk factors of chronic pain in victims of violence was (“risk factor”) AND ((pain)) AND (((“interpersonal violence” OR “domestic abuse” OR “intimate partner violence” OR “partner abuse” OR “violence against women”))). The keywords have been chosen after a preliminary search of the literature, thanks to which it was possible to identify the most used and relevant terms. There were no period restrictions on the search to increase the studies’ yield, though the language was restricted to studies published in English or Italian. Authors were also contacted via email where there was insufficient data, and references from included studies were manually scanned for further sources as per published recommendations [45,46,47]. 

### 2.2. Criteria for Selection of Studies

It included studies on humans of any age with and without a history of abuse during their life identified through published observational study designs (cohort, case–control, and cross-sectional studies). An inclusive approach was adopted for the definition of abuse with a composite of sexual, physical, and psychological violence. Definitions of chronic pain varied between studies and it also adopted an inclusive approach. In general, whatever the type of pain or discomfort, it tends to persist in the involved area for at least 3 of the past 6 months. All bio-psycho-social risk factors were considered. Exclusion criteria included studies without a control group, studies without risk factors, and studies published in non-English or Italian languages. Lastly, systematic reviews, meta-analyses, commentaries, dissertations, thesis, editorials, and conference deeds were excluded but their references were examined to find other studies not retrieved by the search strategy.

### 2.3. Study Selection and Data Extraction

Studies were selected in a three-stage process. All citations identified from initial searching were imported into Zotero Software, where duplicate citations were removed, and after which, two reviewers (AU & CD) independently scrutinized all article titles remaining from the original search. After this, the same two reviewers independently analyzed all article abstracts remaining from the second removal. If there was a disagreement, an independent third reviewer (FF) was consulted. If the abstracts did not provide sufficient information to determine inclusion or exclusion, the reference was included in the next stage (full-text screening) to confirm the information in the full text. Full manuscripts were obtained for studies meeting initial inclusion criteria, and two reviewers (AU & GS) carried out an independent full-text review of all English/Italian language articles. The most recent and complete version of duplicate publications was included in the full-text review, and inadequate versions were excluded. Disagreements regarding inclusion or exclusion criteria were resolved by consensus or consultation of an independent third reviewer (FF). Two reviewers (AU & CD) performed independent data extraction, and where extractable data were missing, authors were contacted by email. They used data to construct tables of risk factors.

### 2.4. Assessment of Study Quality

All studies meeting the selection criteria were assessed for quality based on existing checklists [48]. Quality was defined as the confidence that bias in estimating the effect of risk factors on pain symptom outcomes in victims of violence was minimized through appropriate study design methods and analysis. Two reviewers (AU & GS) independently assessed all studies for quality using predetermined and validated criteria from The Johanna Briggs Institute appraisal checklists for cross-sectional, case–control, and cohort studies [48]. Appraisal criteria included comparability and appropriateness of cases and controls, reliable and valid exposure measurement, identification of confounding factors and whether strategies were implemented to deal with these factors, valid and reliable outcomes assessment, and appropriateness of statistical analyses used. A high-quality study was considered to meet most of these criteria: cross-sectional studies met at least 5/8 criteria, cohort studies fulfilled at least 6/11, and case–control studies met at least 6/10. Low-quality studies were excluded from our review. Cohort studies satisfy 7/11 criteria, 2 case–control studies satisfy 7/10, 2 of them 8/10, 6 cross-sectional studies satisfy 5/8 criteria, 3 satisfy 7/8 criteria, and the last 10 satisfy 6/8 criteria. The utilized appraisal checklists are available as Appendix A.

## 3. Results

### 3.1. Literature Identification, Study Characteristics, and Quality

The search protocol identified 116 publications from online databases. There were 17 removed as they were duplicate publications. The remaining 99 studies were screened against title and abstract criteria, after which 51 were excluded. Of the 48 studies selected for full-text review, 23 were excluded, 3 were reviews, 6 were written in unknown languages, 5 had no pain condition, 5 had pain caused by an injury, and 4 focused on the offender’s group. After, the quality assessment was carried out on 25 studies [49,50,51,52,53,54,55,56,57,58,59,60,61,62,63,64,65,66,67,68,69,70,71,72,73]; see the flow diagram in Figure 1. 

The years of the studies range from 2004 to 2022; 19 studies are cross-sectional, 4 are case–control, and 2 are cohort studies. Of the included studies, 44% are from the United States of America (USA) (11), 12% are from Australia (3), 8% are from Canada (2), and 36% are from other countries (Spain, Thailand, Brazil, Slovenia, Turkey, Serbia, Pakistan, Oman, South Africa). The sample size ranges from 23,846 to 37, age ranges from 15 to 98, and all the abuse and chronic pain types are represented.

### 3.2. Risk Factors

According to the bio-psycho-social model [43] and what was found in the selected studies, there are three specific categories of risk factors: biological, psychological, and sociological. Of these categories, the biological one can be divided into weight conditions, acute upper/lower respiratory tract affection, genitourinary conditions, cardiovascular symptoms and conditions, endocrine disease, hormonal conditions, gastrointestinal disorders, skin problems, and specific inflammations. The psycho-social risk factors can be divided into mental health disease, use of psychoactive substances, life events, life quality, and personal characteristics. Inside the 14 categories, there are many signs, symptoms, and conditions, for a total of 65, as described in Table 1.

### 3.3. The Impact of Risk Factors in the Studies

The selected studies give information about the complexity of the treated problem; in fact, each of them includes both biological and psycho-social aspects. All 25 studies involve psycho-social conditions, and 14 also involve biological conditions.

#### 3.3.1. Biological Risk Factors

Within the biological risk factors, weight conditions are present in 20% of the studies; five are about obesity, and one is about an underweight condition. Acute upper/lower respiratory tract affections are present in 12% of the studies; in three, the specific conditions are asthma, allergic rhinitis, and sinusitis, and in one, it is nasal congestion. Genitourinary conditions are reported in 40% of the studies; in five of them, there are urinary tract infections, in three of them, there are sexually transmitted infections and genital infections, and in one case, there is a vaginal bulge, urinary leakage, genital vesicles, genital ulcers, postcoital bleeding, or prolapse. Cardiovascular symptoms and conditions are reported in 24% of the studies; in five, the condition is hypertension, and in one, it is heart palpitation. Endocrine diseases are present in 12% of the studies; in three, there is diabetes, and in one, there is a disorder of lipid metabolism, thyroid disease, or high cholesterol. In 8% of the studies, they reported a hormonal condition that involved menopausal symptoms one time and dysmenorrhea/irregular menstrual cycle two times. In 20% of the studies, they reported a gastrointestinal disorder that involved irregularities in bowel functioning four times and gastroesophageal reflux one time. Skin problems are present in 8% of the studies; in one, there is dermatitis and eczema, and in the other, there is a rash. Lastly, specific inflammations are reported in 12% of the studies; one time regarding otitis and conjunctivitis, one time regarding muscle inflammation, and one time regarding osteoarthritis.

#### 3.3.2. Psycho-Social Risk Factors

Inside the psycho-social risk factors, mental health diseases are present in 64% of the studies; 13 of them involved a depressive or mood disorder, 9 of them an anxiety disorder, 6 of them a sleep disorder, 4 of them PTSD, 3 a psychosomatic disorder, and in one case, an eating disorder or general history of psychiatric disorder. The use of psychoactive substances is reported in 36% of the studies, presenting drugs and alcohol use in five cases, respectively, and smoking in six cases. Specific life events are noticed in 100% of the studies; in particular, are always-present IPV referred to in 18 studies, childhood abuse, referred to in 3 studies, and both, referred to in 4 studies. Following this, ACEs and parental psychopathology are referred to in two studies, respectively, and parental marital conflict, poor parent–child relationship, abortions, and re-victimization are referred to in one study, respectively. In 24% of the studies, they referred to a life quality condition; in two cases, it concerned life dissatisfaction, mental distress, and suicidal thoughts, and in one case, it concerns feelings of shame and guilt, low self-esteem, sexual dissatisfaction, family/social problems, low social support, reduced physical functioning/physical inactivity. Lastly, 24% of the studies refer to at least one of the following characteristics: role of emotions, tiredness, low vitality, and number of sexual partners in one case, the age of first sexual intercourse/sex too soon in two cases, and painful intercourse in four cases. These results are summarized in Table 2.

## 4. Discussion

This study aimed to explore the bio-psycho-social factors strongly correlated to the onset of chronic pain in violence victims. Therefore, a scoping review was conducted, and now it is possible to make some considerations.

### 4.1. The Most Common Conditions

This scoping review showed a significant trend for the psycho-social aspects of the phenomenon. They are present in all the included studies and have the central frequency inside the categories. Considering 14 categories, of which 9 are biological, and 5 are psycho-social, and life events and mental health diseases have a prevalence of 25% and 16%, respectively, the most significant two. In particular, the abuse conditions (IPV and childhood abuse), adverse experiences, and mood, depressive and anxiety disorders (PTSD included) are the most reported. This evidence is in line with studies that showed a relationship between violence and chronic pain [9,10], and some mental health diagnoses and chronic pain [24,25,26,27,28,29,30,31,32,33]. A particular condition is related to sleeping disorders. There is a literature trend of their involvement, but previous studies do not report a clear and direct relationship with chronic pain or violence. Also in the included studies, they are always present with depression and anxiety disorders comorbidity and never individually. They could be referred to as a secondary diagnosis or a symptom of other conditions, rather than a diagnosis in itself. The nonspecific psychopathology, eating, and psychosomatic disorders would appear less frequent, such as parental marital conflict, poor parent–child relationship, parental psychopathology, and re-victimization. Also, in these cases, the relationship may not be direct but instead increases the odds of the onset of another clinical condition, increasing the odds of the onset of chronic pain.

The genitourinary conditions category has a prevalence of 10%, with a significant trend in urinary tract, genital, and sexually transmitted infections. These could be caused by sexual violence, inducing chronic pelvic pain [75]. Unfortunately, this association is not only for sexually abused but also for the other types. In this case, it is possible to hypothesize a concomitance of biological and psychological aspects involving life quality and personal characteristics components. On the one hand, there are biological infections; on the other hand, maybe there is sexual dissatisfaction, having first sexual intercourse at an age in which there are not adequate cognitive abilities to process the experience, and painful intercourse, which can be both infectious and traumatic. Vaginal bulge, urinary leakage, genital vesicles/ulcers, postcoital bleeding, and prolapse are minor influences in determining chronic pain and can be symptoms or not of the primary diagnoses. In personal characteristics, which have a prevalence of 6%, the only two influential elements are those related to sexual/genital/urinary infections: sex too soon and painful intercourse. The number of sexual partners, emotions, tiredness, and low vitality are not. Tiredness and low vitality may be present later due to chronic pain rather than as an underlying vulnerability.

The use of psychoactive substances has a trend of 9% and is in line with studies that showed a bidirectional relationship with chronic pain [34,35,36]. Their use could interfere with the neural process of restoration of homeostatic conditions and then be used to manage pain.

Life quality and cardiovascular symptoms and conditions have a trend of 6%, with life dissatisfaction, mental distress, suicidal thoughts, hypertension, and heart palpitation as the most referred to. There could be a link between mental distress and cardiovascular conditions; in fact, hypertension and heart palpitations are both influenced by high stress levels [76,77]. Violence causes increased stress [78], which causes hypertension and heart palpitations that predispose to chronic chest pain [19]. Also, suicidal thoughts are influenced by stress [79], and their prevalence is in line with studies that showed a bidirectional relationship with chronic pain [37,38]. Feelings of shame and guilt, low self-esteem, family and social problems, low social support, and physical inactivity are less critical in the life quality category. Low self-esteem is more of a risk factor for IPV than chronic pain in victims [80], and family/social problems and low social support, and feelings of shame and guilt are those referred to after the violence. Instead, physical inactivity is a consequence of pain.

### 4.2. Other Conditions

Weight conditions and gastrointestinal disorders have a trend of 5%; in particular, present are obesity and irregularities in bowel functioning, being less underweight, and gastroesophageal reflux. Clear evidence has shown a link between IPV and obesity [81,82], and PTSD and depression have a crucial mediating role in this [83,84]. The pathway through which the severity of abuse leads to obesity is similar to the mechanism by which obesity and chronic pain exacerbate each other [85]. For the specific irregularities in bowel functioning cases, the evidence is still unclear.

Acute upper/lower respiratory tract affection, endocrine diseases, and specific inflammations have a trend of 3%, and the most reported conditions are asthma and diabetes; allergic rhinitis, sinusitis, nasal congestion, disorder of lipid metabolism, thyroid disease, high cholesterol, and all of the specific inflammations appear to be indirectly related to chronic pain in violence victims. Previous studies found that several factors contribute to the risk of developing asthma, including obesity, female sex, high levels of family stress, and IPV [86,87,88], and experimental studies have yielded novel insight into the potential pathways underlying the connection between these conditions and chronic pain, such as stress-related changes in epigenetic processes, gene expression, and immune responses [89]. Adler [90] has already found a correlation between inflammations and chronic pain, and it is clear that the reason for its presence in this specific sample is probably that it is an a-specific risk factor.

Lastly, skin problems and hormonal conditions have a trend of 2%, with dysmenorrhea/irregular menstrual cycles as the most reported situation. Skin problems, such as dermatitis, eczema, and rash, are related to stress levels, which worsen them [91]. The brain areas related to chronic pain and some psychiatric conditions (e.g., dorsolateral PFC) could significantly influence the skin problem progression, but also, in this case, there is not a direct effect on IPV. John and colleagues [92] and Letourneau and colleagues [93] found an association between violence and dysmenorrhea/irregular menstrual cycles, in particular for sexual or physical violence [94], that increases the risk of the onset of pelvic pain [95], but the reasons of this association need further investigations.

### 4.3. Limitations

This study has some limitations: First, the research method is a scoping review. PRISMA-ScR and PECOS guidelines are used to fill this gap, and a quality assessment was conducted, but other sources of information are excluded, and the grey literature has not been sufficiently analyzed. Then, the non-English/Italian article exclusion leads to the non-inclusion of six potentially relevant articles. Moreover, the inclusion of only observational study designs, due to the absence of interventional studies, may make the results less generalizable. Furthermore, the included studies are not enough to totally clarify the research question. Future directions of this topic will have to increase the number of studies and then conduct a systematic review.

## 5. Conclusions

Intimate partner violence is a health problem that has many physical and psychological consequences, including chronic pain. This study showed that some biological and psycho-social components increase the odds of chronic pain onset. Unfortunately, how they interact with each other is not yet fully established. By continuing the studies in this direction, it will be possible exactly establish the specific and a-specific risk factors and their relationships. After that, it could be possible to create a screening tool to direct women to the correct treatment and individualized treatment that is not possible at the moment because the specific considerable variables were unknown. Surely, future studies could investigate the relationship between biological and psycho-social risk factors to establish cause–effect associations, and to understand which of those found in the trends are relevant in these patients. Despite the limitations, the presenting study is the first to examine the literature trends about factors that accompanied chronic pain in violence victims.

## Figures and Tables

**Figure 1 healthcare-11-02421-f001:**
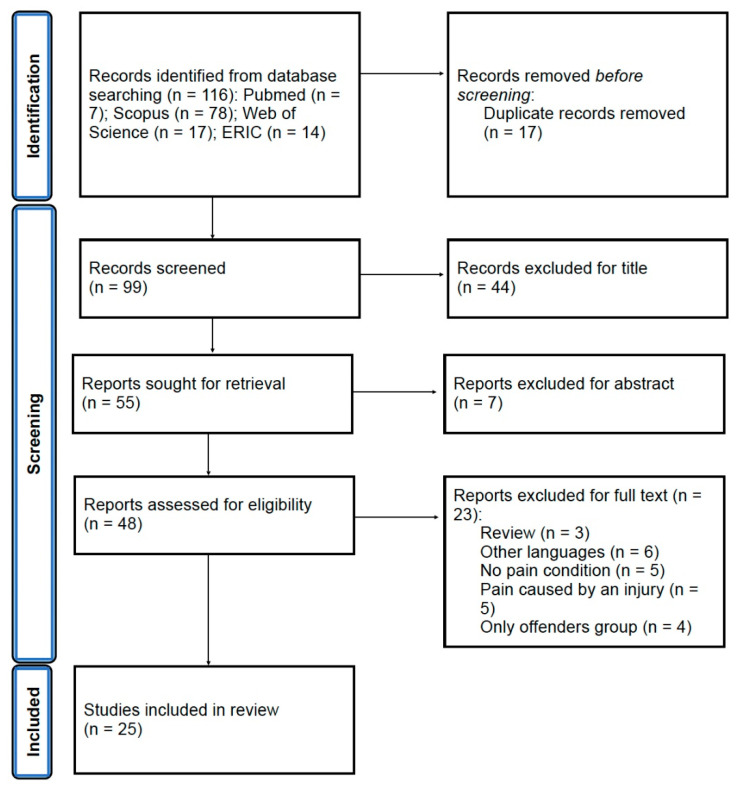
PRISMA flowchart [74].

**Table 1 healthcare-11-02421-t001:** List of risk factors divided into categories.

Biological Risk Factors	Psycho-Social Risk Factors
**Weight condition** Obesity Underweight **Acute upper/lower respiratory tract** **affection** Asthma Allergic rhinitis Sinusitis Nasal congestion **Genitourinary conditions** Sexually transmitted infections (AIDS, chlamydia) Genital infections (vaginitis, vulvitis, cervicitis) Urinary tract infections (cystitis, urethritis) Vaginal bulge Urinary leakage Genital vesicles/ulcers Postcoital bleeding Prolapse **Cardiovascular symptoms and conditions** Hypertension Heart palpitation **Endocrine disease** Disorder of lipid metabolism Diabetes Thyroid disease High cholesterol **Hormonal conditions** Menopausal symptoms Dysmenorrhea/irregular menstrual cycle **Gastrointestinal disorders** Gastroesophageal reflux Irregularities in bowel functioning (diarrohea, constipation, dischezia) **Skin problems** Dermatitis Eczema Rash **Specific inflammations** Otitis Conjunctivitis Muscle inflammation Osteoarthritis	**Mental health disease** Sleeping disorders Anxiety disorders Depressive disorders Mood disorders Post-traumatic stress disorder (PTSD) Psychosomatic disorders Eating disorders History of psychiatric disorders **Use of psychoactive substances** Drugs Alcohol Smoking **Life events** Intimate partner violence (IPV) Childhood abuse Witnessing violence (including in ACEs) Adverse childhood experiences (ACEs) Parental psychopathology Parental marital conflict Poor parent–child relationship Number of abuse (revictimization) Abortion **Life quality** Life dissatisfaction Mental distress Suicidal thoughts Feeling of shame and guilty Low self-esteem Sexual dissatisfaction Family and social problem Low social support Reduced physical functioning/physical inactivity **Personal characteristics** Role emotional Tiredness Low vitality Number of sexual partner Age of first sexual intercourse/sex too soon Painful intercourse

**Table 2 healthcare-11-02421-t002:** Characteristics and results of the included studies.

Authors	Sample	Study Design	Pain	Risk Factors
Ali et al. (2021) Pakistan [69]	945 F 15–49 y/o	Cross-sectional	Pelvic pain	Life events (1): IPV Personal (1): painful intercourse Hormonal (1): dysmenorrhea Genitourinary (7): urinary tract infections, genital infections, urinary leakage, genital vescicles/ulcers, postcoital bleeding, and prolapse
Al Kendi et al. (2021) Oman [70]	978 F 30.6 y/o	Cross-sectional	General chronic pain	Life events (1): IPV Mental health (3): depression, sleeping, and psychosomatic disorders
Bonomi et al. (2007) USA [50]	1928 F 18–64 y/o	Cross-sectional	General chronic pain	Life events (1): IPV Life quality (1): family and social problems Mental health (3): sleeping, anxiety, and depressive disorders Substances (2): smoking and drugs Respiratory (3): allergic rhinitis, asthma, and sinusitis Cardiovascular (1): hypertension Endocrine (3): disorder of lipid metabolism, thyroid disease, and diabetes Inflammations (2): otitis and conjunctivitis Genitourinary (2): genital infections and urinary tract infections Hormonal (2): menopausal symptoms and irregular menstrual cycle Gastrointestinal (1): gastroesophageal reflux Skin (2): dermatitis and eczema Weight: obesity
Chartier et al. (2010) Canada [52]	9953 (5187 F—4766 M) 15–98 y/o	Cross-sectional	General chronic pain	Life events (4): childhood abuse, parental marital conflict, parental psychopathology, and poor parent–child relationship
De Wet-Billings & Godongwana (2021) South Africa [71]	216 F 15–34 y/o	Cross-sectional	General chronic pain	Life events (2): IPV Cardiovascular (1): hypertension
England-Mason et al. (2018) Canada [64]	23,846 (12,290 F—11,556 M) 18–64 y/o	Cross-sectional	General chronic pain	Life events (2): childhood abuse and witnessing violence Mental health (2): mood and anxiety disorders Substances (2): smoking and drugs
Eslick et al. (2011) Australia [55]	87 (66 F—21 M) 47 y/o	Case-control	General chronic pain	Life events (1): childhood abuse Mental health (1): depressive disorder
FitzPatrick et al. (2022) Australia [72]	1507 F 31 y/o	Cohort	Pelvic pain	Life events (1): IPV Mental health (2): anxiety and depression disorders Personal (2): age of first sexual intercourse and painful intercourse Genitourinary (1): urinary leakage Weight: obesity
Gelaye et al. (2016) USA [61]	2970 F 28.1 y/o	Cross-sectional	General chronic pain	Life events (3): IPV, childhood abuse, and number of abuse
Gerber et al. (2017) USA [58]	92 F 39 y/o	Cross-sectional	General chronic pain	Life events (1): IPV Mental health (1): PTSD Substances (2): smoking and alcohol
Grossi et al. (2018) Brazil [65]	80 F 33 y/o	Case-control	General chronic pain	Life events (1): IPV Mental health (2): depression and psychosomatic disorders
Gucek & Selic (2018) Slovenia [66]	161 F 51.1 y/o	Cross-sectional	General chronic pain	Life events (1): IPV Mental health (5): depression, anxiety, sleeping, eating, and psychosomatic disorders Life quality (3): reduced physical functioning, feelings of shame and guilty, and low self-esteem Substances (1): smoking Inflammations (1): muscle Gastrointestinal (1): irregularities in bowel functioning Genitourinary (2): genital infections and urinary tract infections
Gunduz et al. (2019) Turkey [67]	136 F 40 y/o	Case-control	Fibromyalgia	Life events (2): IPV and parental psychopathology Mental health (4): history of psychiatric disorder, PTSD, mood, and anxiety disorders Substances (2): smoking and alcohol
Halpern et al. (2017) USA [62]	37 F 19–63 y/o	Cross-sectional	General chronic pain	Life events (1): IPV Mental health (2): PTSD and anxiety disorder Cardiovascular (2): heart palpitation, hypertension
Hegarty et al. (2008) Australia [51]	942 F 16–50 y/o	Cross-sectional	General chronic pain	Life events (1): IPV Mental health (3): depression, anxiety, and sleeping disorders Personal (1): tiredness Life quality (1): suicidal thoughts Genitourinary (1): urinary leakage Gastrointestinal (1): irregularities in bowel functioning Respiratory (1): nasal congestion Skin (1): rash Cardiovascular (1): hypertension
Jovanovic et al. (2020) Serbia [68]	6320 F 20–75 y/o	Cross-sectional	General chronic pain	Life events (2): IPV and abortions Mental health (2): depression and sleeping disorders Substances (1): alcohol
Kelly et al. (2011) USA [56]	135 F 40.3 y/o	Cross-sectional	General chronic pain	Life events (2): childhood abuse and IPV Mental health (4): PTSD, sleeping, depression, and anxiety disorders Life quality (2): life dissatisfaction and suicidal thoughts
Lutgendorf et al. (2017) USA [63]	188 F 18–64 y/o	Cross-sectional	Pelvic pain	Life events (1): IPV Personal (1): painful intercourse Genitourinary (1): vaginal bulge Gastrointestinal (1): irregularities in bowel functioning
Parish et al. (2004) USA [49]	3323 (1662 F—1661 M) 20–64 y/o	Cross-sectional	General chronic pain	Life events (1): IPV Life quality (3): life, mental, and sexual dissatisfaction Genitourinary (1): sexually transmitted infections
Raphael et al. (2022) USA [73]	1974 F 60.2 y/o	Cross-sectional	Pelvic pain	Life events (1): IPV Mental health (1): depression disorders Genitourinary (1): urinary tract infections Endocrine (1): diabetes Weight: obesity and underweight
Saito et al. (2013) Thailand [59]	421 F 25.9 y/o	Cross-sectional	General chronic pain	Life events (1): IPV Personal (2): role emotional and low vitality Mental health (1): depressive disorder
Sutherland et al. (2013) USA [60]	145 F 30.1 y/o	Cross-sectional	Pelvic pain	Life events (2): IPV and childhood abuse Personal (3): number of sexual partner, painful intercourse, and age of first sexual intercourse Substances (2): alcohol and drugs Genitourinary (2): sexually transmitted infections and urinary tract infections
Vives-Cases et al. (2010) Spain [53]	13,094 F 16–64 y/o	Cross-sectional	General chronic pain	Life events (1): IPV Life quality (2): low social support and mental distress Substances (2): smoking and drugs Cardiovascular (1): hypertension Weight: obesity
Williams et al. (2010) USA [54]	309 F 18–64 y/o	Case-control	Pelvic pain	Life events (2): childhood abuse and IPV Genitourinary (1): sexually transmitted infections
Young et al. (2011) USA [57]	360 (260 F—100 M) 53 y/o	Cohort	General chronic pain	Life events (2): IPV and ACEs Mental health (2): depression and anxiety disorders Substances (2): alcohol and drugs Inflammations (1): osteoarthritis Respiratory (1): asthma Endocrine (2): diabetes and high cholesterol Gastrointestinal (1): irregularities in bowel functioning Cardiovascular (2): hypertension and hearth palpitation Weight: obesity

## Data Availability

The data reported here, and the quality assessment tables and checklists are available upon request.

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
