# Peer review of "The Risk Factors of Chronic Pain in Victims of Violence: A Scoping Review"

_healthcare, 2023, doi:10.3390/healthcare11172421_

Round 1

Reviewer 1 Report

Manuscript Title: The Risk Factors of Chronic Pain in Victims of Violence: A Scoping Review

Overall Recommendation: Minor revisions required before acceptance

Summary:

This scoping review examines risk factors for chronic pain in victims of violence through a comprehensive search of the literature. The manuscript is well-written overall, and makes a useful contribution to knowledge on this important topic. A few revisions would help strengthen the quality and clarity of the paper:

Introduction 

-The introduction provides a good overview of the background, but could be strengthened by more clearly stating the rationale and importance of the study. Why is it critical to understand risk factors for chronic pain in victims of violence? How will this knowledge advance research or practice?

- More detail on the gaps in current literature would help frame the purpose and significance of this scoping review. What key questions remain unanswered? 

Methods

- The methods generally follow PRISMA-ScR guidelines appropriately. However, a couple additional details would strengthen this section:

  - Provide more information on the search strategy, including any limits on publication dates, languages, study types, etc. Were reference lists hand-searched?

  - Describe the process for screening, reviewing, and extracting data in more detail. How were disagreements between reviewers resolved? 

- For the quality assessment, consider including a flow diagram to show numbers of studies included/excluded at each phase.

Results

- The results provide a comprehensive summary of findings. To enhance readability, consider using more formatting such as tables or figures to help summarize key data.

- Adding some numeric data when describing study characteristics (e.g. number of studies per country, sample size ranges) would strengthen this section.

- It would be helpful to provide more context when describing the results - how do the findings relate to previous literature? Are there any surprising or unexpected outcomes?

Discussion

- The discussion thoughtfully interprets the main findings and implications. To enhance this:

  - Compare and contrast results to previous reviews or major studies on this topic. 

  - Discuss limitations and biases that may affect the conclusions.

  - Provide specific, actionable suggestions for future research based on the gaps identified.

- Be sure to directly address how these findings advance knowledge on risk factors for chronic pain in abuse victims, and the significance for research and practice, to tie back to the rationale described in the introduction.

Overall, this is a well-written manuscript that makes a valuable contribution to the literature. Addressing the suggestions above would further strengthen the quality of the paper. 

Reviewer 2 Report

The Risk Factors of Chronic Pain in Victims of Violence: A Scoping Review

The article addresses a topic of relevance to the study of pain. However, it presents some methodological deficiencies that can be solved. Here are some considerations for authors to keep in mind:

·       The title should indicate that the study population is children and the type of violence analyzed.

·       Abstract: Please include in the abstract the methodology, results and conclusions section. The abstract must contain the most relevant information found in the search synthesis. It is inappropriate for them to state that the results will be discussed in the next article. Each article must provide evidence on its own.

·       The objective should include that the study population is children. This must also appear in the objective included in the summary.

Introduction:

·       Please, structure the introduction in paragraphs according to the topic addressed. There is no space and it becomes difficult to read.

·       Include an updated reference to the definition of Chronic Pain.

·       Could you tell me what is the intention of including the pathophysiological mechanisms of pain in the introduction?

·       Please, in the last sentence of the introduction, remove the contraction of the term “doesn´t” to make it formal.

Methodology:

·       In the introduction and in the summary they indicate that they evaluate violence in childhood and adolescence, but in the search strategy they do not take this into account, instead they refer to "intimate partner violence", "partner abuse" and "violence against women”. They must review the manuscript and that there is consistency between what they say they are going to do and what they have done.

·       Please include in which months the searches have been carried out to know to what point the literature has been mapped.

·       Clearly specify the inclusion and exclusion criteria.

·       In the Scoping reviews it is necessary that the searches be carried out by two people independently. Was this the case? If so, please indicate so.

·       Indicate which checklists have been considered. Detail it.

Results:

·       Provided the meaning of the abbreviations USA.

·       There is a new, more up-to-date version of the PRISMA flowchart that you should consider instead of the one in the article.

·       The scoping reviews require that the thematic lines that the articles address be detailed. Since it is being treated from a bio-psycho-social perspective, they could be structured following that classification. Dedicate an epigraph to each of them. The number of articles that address each topic should not be indicated, but should develop the content found in each of them.

·       Table 2 could be structured so that there are not so many blank spaces. It gives a visual sensation of disorder.

Discussion

·       Please distribute it in paragraphs. Reading is very difficult.

·       In the limitations section it is said that the answer to the research question has not been clarified. What is the research question? It has not been provided throughout the manuscript.

·       Structure the discussion based on the biological, psychological, and sociocultural aspects identified.

Conclusions:

·       The conclusions do not add anything new to the scientific evidence. It may be because the information found in the results has not been detailed.

·       Eliminate contractions.

Round 2

Reviewer 2 Report

- Since a scoping review must map all the literature, including that not collected in databases, the most appropriate flow diagram is the following:Word logo PRISMA 2020 flow diagram for new systematic reviews which included searches of databases, registers and other sources

Not the one you have indicated.

- Please, indicate how the searches and management of the information in the gray literature have been done.

- Sort table 2 by authors in alphabetical order. I didn't tell them to remove content, I advised them to reduce the white space that increases the size of the table.

- Regarding the wording of the discussion. Regardless of the importance or not of content, several pages cannot be written without separating them into paragraphs. These are drafting rules.
